# Measuring the Air Quality Using Low-Cost Air Sensors in a Parking Garage at University of Minnesota, USA

**DOI:** 10.3390/ijerph192215223

**Published:** 2022-11-18

**Authors:** Andres Gonzalez, Adam Boies, Jacob Swanson, David Kittelson

**Affiliations:** 1Department of Civil, Environmental, and Geo-Engineering, University of Minnesota, Minneapolis, MN 55455, USA; 2Department of Engineering, University of Cambridge, Cambridge CB2 1PZ, UK; 3Department of Integrated Engineering, Minnesota State University, Mankato, MN 56001, USA; 4Department of Mechanical Engineering, University of Minnesota, Minneapolis, MN 55455, USA

**Keywords:** underground parking garage, low-cost sensors, air pollution, air pollutants, particles, gases, fuel emissions

## Abstract

The concentration of air pollutants in underground parking garages has been found to be higher compared to ambient air. Vehicle emissions from cold starts are the main sources of air pollution in underground parking garages. Eight days of measurements, using low-cost air sensors, were conducted at one underground parking garage at the University of Minnesota, Minneapolis. The CO, NO, NO_2_, and PM2.5 daily average concentrations in the parking garage were measured to be higher, by up to more than an order of magnitude, compared to the ambient concentration. There is positive correlation between exit traffic flow and the air concentrations in the parking garage for lung deposited surface area (LDSA), CO_2_, NO, and CO. Fuel specific emission factors were calculated for CO, NO, and NOx. Ranging from 25 to 28 g/kg_fuel_ for CO, from 1.3 to 1.7 g/kg_fuel_ for NO, and from 2.1 to 2.7 g/kg_fuel_ for NOx. Regulated emissions were also calculated for CO and NOx with values of 2.4 to 2.9 and 0.19 to 0.25 g/mile, respectively. These emissions are about 50% higher than the 2017 U.S. emission standards for CO and nearly an order magnitude higher for NOx.

## 1. Introduction

The world automobile production increased from 30 million in 1983 to 56.6 million in 2016 while the U.S. increased the number of automobiles per 1000 people from 724 to 831 during the same period [1]. The breakdown by state shows that Minnesota has 963 automobiles per 1000 people [2]. The increase of automobile use and space constraints imply growing demand for parking garages [3,4]. Other studies show that automobile use increases where more parking spaces are available [5,6,7]. There is a strong correlation between parking supply and automobile use in cities [6,8]. 

Many studies have shown that the concentrations of air pollutants in parking garages are higher compared to in the urban air. This will be discussed in more detail below along with the results of our measurements. There are three causes that explain the higher concentrations. First, the higher number and flow of vehicles [4]; therefore, there are differences between weekdays and weekends [9]. Second, emissions are higher during the vehicle’s cold start mode [3,10]. Finally, there is an absence of efficient ventilation systems in many parking garages [11]. Gaseous pollutant concentration in parking garages might be affected by season and environmental conditions as well [12]. 

There are several studies measuring and modelling the air quality in parking garages and its health effects [13]. Parking garage workers are exposed for long periods of time to traffic-related air pollution [14]. People who have exposure to high concentrations of air pollutants in the short-term, such as drivers and passengers, may experience acute coronary events [15]. There are serious health hazards for drivers and passengers even if they stay in the parking garages for short periods of time [16]. Diesel exhaust air pollution exposures have shown microglia activation, increased lipid peroxidation, and neuro-inflammation in various brain regions [17]. An increase of 10 µg m^−3^ of PM2.5 is associated with a 3.4% increase in daily mortality [18]. PM2.5 exposure to concentrations in the range of 100–200 µg m^−3^ provoke alveolar inflammation, with release of mediators capable, in susceptible individuals, of causing exacerbations of lung disease and of increasing blood coagulability [19]. Long-term exposures to PM2.5 are associated with increased risks of all-cause and cardiopulmonary mortality [20]. 

The health effects of exposure to CO at low-concentrations are cardiovascular and neurobehavioral while at high-concentrations unconsciousness and death may occur after acute or chronic exposure [21]. CO can reach the placenta, affecting circulation to the fetus and its developing brain [22]. Cardiovascular effects are observed at CO exposure in the range of 10–15 ppm [23]. NO_2_ exposure causes adverse respiratory health effects, and in spaces with low air-circulation it may cause severe lung injury and death [24]. An increase of 5 ppb of NO_2_ 24 h-average causes an increase of 6% in the asthma related hospital admissions for children between 5–14 years [25]. One-hour exposure of ~25 ppb NO_2_ is associated with 1.3% increase in the daily number of deaths [26]. The effects of high-level CO_2_ exposure are physiologic, toxic, and potentially lethal [27]. CO_2_ levels should not exceed 1000 ppm [28]. 

To develop strategies to control air pollution it is critical to have reliable air quality data. Stationary air monitoring stations help local authorities and environmental agencies in achieving this goal. However, these measurements have limitations, air monitoring stations provide temporal data on air quality, but only at discrete locations at relatively high cost [29]. An alternative method, low-cost air monitoring sensors, can complement air monitoring stations. Low-cost air pollution sensors can cover larger areas and the costs of typical sensors are $150–200 each [29]. However, data quality is the main concern for their measurements [30]. The periodic re-calibration of the low-cost sensor can improve the accuracy, data quality and reliability [31,32]. The major error sources and recalibration strategies suited the low-cost sensor for different deployments are discussed by Concas [32].

The aim of this study is to measure the air quality in a parking garage at the University of Minnesota East-Bank campus. The measurements were conducted during an eight-day period using low-cost, mobile air quality monitoring (LCMAQM) sensors. The gases measured were carbon monoxide (CO), carbon dioxide (CO_2_), nitrogen dioxide (NO_2_), nitric oxide (NO), and ozone (O_3_). The particles measured were PM2.5 and lung deposited surface area (LDSA). A wireless Mobile Autonomous Air Quality Sensor box (MAAQSBox) contains LCMAQM sensors (gas and particle) and a wireless broadcasting system. Traffic flow was also measured. Daily and weekday versus weekend trends are examined. The results will be also compared to ambient air quality measured by the air monitoring stations (AMS) nearby. 

## 2. Materials and Methods

### 2.1. Location

The MAAQSbox was placed inside of the Church Street Garage (CSG) on East Bank at the Minneapolis campus of the University of Minnesota (UMN). The MAAQSbox was on the third below ground level among six levels as it shown in Figure 1a,b. The CSG is located at 80 Church St SE; this is shown Figure 1c. The CSG presents visitor/public parking hourly rates and contract parking. The parking garage has 237 spaces, and the users are UMN employees, professors, students, and visitors. All the spaces are located underground. According to Transportation Department of UMN there is no significant difference in terms of traffic between regular semester and summer. 

### 2.2. Sensor Technology

#### 2.2.1. Description of MAAQSbox

The MAAQSBox is an autonomous device that houses five gas sensors and two particle sensors, includes thermal conditioning of sample streams and continuously broadcasts measured concentrations [33,34]. The gas sensors are CO, O_3_, NO_2_, and NO produced by AlphaSense. Each sensor contains four sections: Working (We), Auxiliary (Ae), Reference, and Counter electrodes [35]. The B4 sensors detection limits range from 2 to 10 ppb [36]. The CO_2_ gas sensor uses non-dispersive infra-red absorption technology produced by Yoctopuse. The CO_2_ gas sensor measuring range 0–100,000 ppm, and the accuracy and sensitivity are 30 and 20 ppm, respectively [37]. All the gas sensors were installed in a Flow Sensing Cell Apparatus (FSCA) [33,34]. The MAAQSBox includes two particle sensors [33,34]. The first is OPC-N2, produced by AlphaSense. The second particle sensor is Partector and it was produced by Naneos. This sensor measures particles in a size range from 10 nm to 10 µm weighted to approximate the product of particle surface area and inhaled deposition fraction in the alveolar region of the respiratory system. This is called the lung deposited surface area (LDSA). The Partector concentration ranges from 1–20,000 µm^2^/cm^3^ and size from 10 nm to 10 µm [38]. Due to low performance during the calibration the OPC-N2 is not included in this study.

#### 2.2.2. External Particle Sensor

The TSI SidePak Personal Aerosol Monitor AM510 is a laser photometer (TSI 2019) used to measure the mass concentration of particles smaller than 2.5 µm. The aerosol stream passes into the optical chamber, where they are illuminated with a beam of laser light [39]. The intensity of scattered light is proportional to the PM2.5 concentrations [40]. The particle size range from 0.1 µm to 10 µm and the minimum resolution is 0.001 mg/m^3^ [41].

#### 2.2.3. MAAQSBox Field Calibration

Table 1 shows the results of the sensor calibrations. The aim of the field calibration was to assess LCMAQM sensor performance compared to reference instruments in the field [33,34]. Thus, the MAAQSBox was installed next to an AMS. The field calibration was conducted for CO, NO, NO_2_, and O_3_ gas sensors [33,34]. The calibrations of LCMAQM sensors were determined by multivariate linear regressions (MLR). The fit model improved including the temperature and humidity [33,34]. The field calibration was conducted between 407 to 515 h (Table 1) depending of each sensor. Appendix A shows the temperature and humidity from the sensors and Partector. The Partector, SidePak, and the CO_2_ sensor were run as received from the manufacturer using factory calibration. Table 1 shows calibration results of B4 gas sensors. N represents the number of hours of calibration for each sensor in the AMS. R^2^ is the coefficient of determination of MLR. SE is the standard deviation of mean. RC represents the range of the concentration during the calibration.

#### 2.2.4. Measurements and Statistical Analysis

The measurements at CSG were conducted for eight days, between 12:00 a.m. 24 July and 12:00 p.m. 31 July. The data analysis was conducted in Excel. The data were averaged and recorded every 1 min. Data are analyzed and presented in four ways. First, we examine the correlation between the pollutants measured and entry and exit traffic. Here, the entry and exit traffic are defined as total vehicles entering and leaving the parking garage every 30 min. Second, 24 h daily average concentrations measured in the parking garage are compared to those measured at the AMS. The third analysis focused on the weekday average variation of pollutant concentration in the garage with time of day and the overall range of 1 h average concentrations in the garage and at the AMS. Lastly, CO_2_, CO, NO, and NO_2_ data are used to calculate fuel specific emissions (pollutant/kg_fuel_) for CO, NO, and NOx (NO + NO_2_) and regulated emissions (g/mile) for CO and NOx.

Due to retrieval issues, data are not available for the CO_2_ sensor between Sunday 11 a.m. and Monday 8 a.m., and between Friday 7 p.m. and Saturday 12 p.m. For the same reason, data are not available for the NO sensor between the first Wednesday 12 a.m. and Thursday 5 p.m. The O_3_ and NO_2_ sensors malfunctioned on the first Wednesday and these data have been excluded.

## 3. Results and Discussion

### 3.1. Parking Garage Air Quality and Traffic

Figure 2 shows daily traffic by day. The total traffic during the eight days of measurement, was 1999 vehicles. The average vehicles per day during weekdays was 322 while during the weekend it was 30. Table 2 shows the correlations among all measured variables. All the correlations are significant (*p*-value < 0.05) exept for NO_2_ and PM2.5 and traffic in and PM2.5 (Appendix A). Pollution levels are much more strongly correlated with exit traffic than entry traffic because of higher emissions during warmup [3]. However, CO_2_ shows a modest correlation with entry traffic as well. This is to be expected. CO_2_ is a marker of hydrocarbon combustion and is emitted by both cold and fully warmed up engines. Engine friction is higher for a cold engine, so fuel consumption and CO_2_ emissions are increased during cold starts, but the increase is modest, ~30% [42] compared to, for example, 400 to 800% increases for hydrocarbons and CO, respectively [43].

Daily 24 h average CO_2_ concentrations are shown in Figure 3a. Due to instrument malfunctions, 24 h daily averages are only available for Tuesdays, Wednesday, and Thursday. The average CO_2_ concentration during these three weekdays is 585 ppm. The AMS did not measure CO_2_ so no ambient values are shown. In Figure 3b 30 min weekday average CO_2_ concentration and entry, exit, and total traffic are plotted against time of day. The concentration increases rapidly in the morning along with entry traffic, continues to grow through early afternoon as exit traffic grows and then falls with decreasing traffic. The highest average CO_2_ concentration of 920 ppm occurs between 2:00 and 2:30 p.m. There are two peaks in average total traffic, 48.5 vehicles (mainly entering) in the morning between 8:30 and 9:00 a.m. and 34.7 vehicles (mainly exiting) in the afternoon between 4:00 and 4:30 p.m. Overnight, between 12:00 and 7:00 a.m. CO_2_ average concentration is 414 ppm, very close to the worldwide average ambient CO_2_ value for 2019 of 411 ppm [44]. Table 3 summarizes the overall 1 h average concentration ranges for all the measured pollutants. For CO_2_, measured concentrations range from 390 to 1270 ppm. This is in the range of other studies [45,46].

Figure 3c shows 24 h daily average CO concentrations. The average CO concentration in the garage during weekdays is 3.17 ppm while during the weekend, when traffic is much lower, it is only 0.68 ppm, nearly a factor of 5 lower. On Thursday, 25 July the parking garage presents the highest daily average concentration with 3.4 ppm. Both weekday and weekend AMS concentrations are much lower, ranging from about 0.4 to 0.5 ppm.

In Figure 3d weekday 30 min average CO_2_, CO, and exit traffic are plotted against time of day. The CO concentration tracks more closely with exit traffic than CO_2_ because of the association of CO with cold starts. The peak CO concentration of 9.2 ppm occurs between 2:00 and 2:30 p.m., somewhat earlier than the exit traffic peak but following the same general trend. These results are similar to other studies in CO patterns and peak times [45,47,48]. Table 3 shows the overall range of 1 h average CO concentrations to be from 0.2 to 11.7 ppm in the garage and from 0.3 to 0.7 ppm at the AMS. The EPA 1 h and 8 h CO standards are 35 and 9 ppm, respectively. The concentrations in the garage are well below these standards.

Other research presented a range of CO concentrations in underground parking garages. Concentrations ranging between 0.44 ppm and 1.42 ppm were reported by Debia [12] while Yang [49] concentration peaks between 11:00 a.m. and 1:00 p.m., and between 3:00 and 6:00 p.m. of 3.24 and 4.86 ppm, respectively. Rukaibi [50] reported values higher than 9 ppm while Demir [51] reported an average concentration of 10.6 ppm with a maximum of 25 ppm in a parking garage in Istanbul, Turkey. Even higher concentrations were reported for a garage in Guangzhou, China ranging from 3 to 69 ppm, which were 48% higher than only local outdoor concentrations [4].

Figure 3e shows weekday and weekend daily average NO concentrations in the garage and ambient levels measured at the AMS. The average NO concentration in parking garage during weekdays is 121 ppb while during the weekend it is 21 ppb, only 17% of the weekday level. Ambient levels are much lower, averaging 6 ppb on weekdays and 2 ppb on the weekend. On Tuesday 30 July both the parking garage and the AMS present the highest daily average concentrations with 140 ppb and 11.2 ppb, respectively. There is no EPA NO ambient standard.

Figure 3f shows NO, CO, and exit traffic. Table 2 shows that CO and NO are strongly correlated, r = 0.96, and this is clear in the figure. Both CO and NO are associated with cold starts and thus exit traffic. The peak NO concentration of 408 ppb occurs between 2:30 and 3:00 p.m., slightly later than the CO peak. The overall range of 1 h average NO concentrations is from 12 to 520 ppb in the garage and from 0.1 to 27 ppb at the AMS. Other research presented a similar NO concentration in an underground parking garage reaching peaks at 11:00 a.m. and 5:00 p.m. of 260 and 330 ppb, respectively [49].

Figure 3g shows weekday and weekend NO_2_ (nitrogen dioxide) concentrations in the garage and ambient levels measured at the AMS. For NO_2_ there is much less difference between indoor and outdoor and weekday and weekend concentrations than for CO and NO. The average NO_2_ concentration in the parking garage during weekdays is 17 ppb while during the weekend it is 13 ppb, 24% lower than during weekdays. Ambient concentrations average 9 ppb weekdays and 7 ppb weekends. The highest daily average concentration in the garage is 19 ppb on Thursday 25 July while that highest measured at the AMS is 15 ppb on Tuesday 31 July. The EPA 1 h NO_2_ standard is 100 ppb while the annual standard is 53 ppb.

Figure 3h shows NO_2_ and NO concentrations and exit traffic. The concentration of NO_2_ is much lower than that of NO and only weakly associated with traffic. Concentrations of NO_2_ are mainly controlled by atmospheric chemistry [52] and direct tailpipe emissions of NO_2_ are expected to be much lower than those of NO [53]. Consequently, NO_2_ is much less dependent upon local traffic emissions. The peak NO_2_ concentration of 21.3 ppb occurs later in the afternoon between 5:30 and 6:30 p.m. The overall range of 1 h average NO_2_ concentrations is from 8.4 to 24.5 ppb in the garage and from 1.0 to 20.0 ppb at the AMS. These values are well below the 100-ppb hourly EPA standard.

Other research in underground parking garages show somewhat lower NO_2_ concentrations with peaks at 11:00 a.m. and 5:00 p.m. of 5 and 16 ppb, respectively [49], and values ranging from 17 to 20 ppb [21,54]. In New Delhi, India the maximum 8 h average concentrations in three parking garages ranged from 14–24 ppb [55]. On the other hand, very high levels were observed in a study in Paris, France, with maximum NO_2_ 1 h averages ranging from 189 to 261 ppb in three parking garages [56].

Daily O_3_ (ozone) concentrations are shown in Figure 3i. Ozone is not a tailpipe emission and O_3_ levels in the garage tend to track with outdoor AMS levels with little difference between weekday and weekend. In fact, the overall average O_3_ concentration in the parking garage is 25.5 ppb close to the average outdoor concentration measured at the AMS of 24.4 ppb. On Sunday 24 July the parking garage presents the highest daily average concentration with 29 ppb, while in AMS the highest is on Friday 26 July with 38 ppb. The EPA 8 h ambient standard is 70 ppb.

Figure 3j shows O_3_ and NOx (NO + NO_2_) concentrations and exit traffic. The O_3_ concentration drops significantly during peak exit traffic and high NOx concentrations. Ozone is not emitted by vehicles and is, in fact, destroyed by vehicle exhaust constituents, especially NO and NO_2_ [57]. The source of O_3_ is outside air. The minimum ozone concentration of 15.3 ppb occurs between 1:30 and 2:00 p.m. concurrent with the highest NOx. This is caused by local titration with NOx (dominated by NO): NO + O_3_ → NO_2_ + O_2_. In the late afternoon and overnight concentrations are in the range of 24 to 27 ppb, presumably associated with ambient air. As shown in Table 3, the overall range of 1 h average O_3_ concentrations is from 7.2 to 35.4 ppb in the garage and from 1.0 to 52.0 ppb at the AMS. The concentrations are well below the 70 ppb 8 h standard.

Figure 4a shows daily average PM2.5 concentrations. The average PM2.5 concentrations in the parking garage during weekdays and weekends are 15.3 and 14.1 µg m^−3^, respectively, while corresponding ambient levels are lower, 6.1 and 5.3 µg m^−3^, respectively. However, daily average concentrations in the garage are strongly correlated with ambient levels with a correlation coefficient of 0.92. Friday 26 July shows the highest daily average concentrations for both the parking garage and the AMS with values of 20.7 and 8.6 µg m^−3^, respectively. The EPA 24 h PM2.5 standard is 35 µg m^−3^ and the daily average limit recommended by WHO is 25 µg m^−3^. Both the highest parking garage and AMS concentrations fall below these standards. Weekday average particle concentrations, PM2.5 and LDSA, as well as exit traffic counts are plotted against time of day in Figure 4b. It is clear from the figure and from Table 2 that PM2.5 is only weakly correlated with exit traffic and CO_2_. On the other hand, LDSA is moderately correlated with these variables and well correlated the main gaseous tailpipe emissions CO and NO (r = 0.75 in both cases). The weekday average concentration of PM2.5 reaches a peak of value 19 µg m^−3^ between 4:30 and 5:00 p.m. As shown in Table 3 the overall range of 1 h average PM2.5 is from 4.2 to 35.5 µg m^−3^ in the garage and from ~0 to 22 µg m^-3^ at the AMS.

Other research on PM2.5 showed 1.54 times higher concentration in an underground garage compared to outdoor concentrations [46,58]. Concentrations of 43 to 60 µg m^−3^ were reported in a garage while at the same time in the street they ranged 23 to 27 µg m^−3^ [14]. In a study of 8 naturally ventilated residential underground parking garages daily average PM2.5 concentrations ranging from 50 to 200 µg m^−3^ were reported by Liu [59]. PM2.5 8 h daily average concentrations in an underground parking facility in India ranging from about 60 to 240 µg m^−3^ were measured by Samal [13].

LDSA is a measure of ultrafine particles which are less stable and more impacted by local conditions than PM2.5. The LDSA concentration follows exit traffic, with a peak value of 49.9 µm^2^/cm^3^ between 4:30 and 5:00 p.m. The overall range of 1 h average LDSA concentrations is from 5.8 to 98.5 µm^2^/cm^3^ in the garage. Kuuluvainen [60] have reported outdoor measurements of LDSA in Finland, with average background and near road values of 12 and 94 µm^2^/cm^3^, respectively. There are no ambient standards for LDSA. Particle emissions from vehicles are not only from the engine but also from tires, brakes, and the vehicle chassis [54]. These tire and brake particles contain copper, barium, iron, and antimony, and other materials [61]. The concentrations of these non-exhaust particles in underground garages may be two orders of magnitude higher compared to outdoor air in urban areas [11]. LDSA is a mainly measure of ultrafine particles [60] found in fresh tailpipe emissions. On the other hand, PM2.5 is influenced not only by tailpipe, but also by tire, brake, and chassis emissions and background levels.

### 3.2. Vehicle Emission Factors

Emissions of CO and NOx have been calculated in fuel specific form and then converted to equivalent emissions in g/mile. Fuel specific emissions were calculated based on the concentrations of CO, NO, NO_2_, and CO_2,_ for example, for CO we have [62]:$$\frac{\mathrm{kg}\;\mathrm{CO}}{{\mathrm{kg}\;\mathrm{CO}}_{2}}=\frac{\left(\;\mathrm{Pollutant}\;\mathrm{added}\;\mathrm{by}\;\mathrm{traffic}\;\frac{\mathrm{kg}}{m^{3}}\right)}{\left({\mathrm{CO}}_{2}\;\mathrm{added}\;\mathrm{by}\;\mathrm{traffic}\;\frac{\mathrm{kg}}{m^{3}}\right)}$$
$$\frac{\mathrm{kg}\;\mathrm{CO}}{{\mathrm{kg}\;\mathrm{CO}}_{2}}*\frac{{\mathrm{kg}\;\mathrm{CO}}_{2}}{\mathrm{kg}\;C}*\frac{\;\mathrm{kg}\;C}{\mathrm{kg}\;\mathrm{Fuel}}=\frac{\mathrm{kg}\;\mathrm{CO}}{\mathrm{kg}\;\mathrm{Fuel}}*\frac{1000g}{\mathrm{kg}}=\frac{\;g\;\mathrm{CO}}{\mathrm{kg}\;\mathrm{Fuel}}$$

The ratio of the pollutant added to CO_2_ added is based on the slope of the regression line of the pollutant concentration against CO_2_ concentration. Similar calculations were done for NO and NOx. Fuel specific emissions were converted to g/mile by multiplying g/kg_fuel_ by estimated kg_fuel_/mile. Assumptions are listed in Table 4 below.

Calculating these emissions was complicated by instrument problems that led to incomplete data sets for CO_2_ and NOx. This is illustrated in Figure 5. The NO sensor did not work during the first day and most of the second day and the CO_2_ sensor was intermittent over the weekend. As a result, full day (24 h) comparisons are only possible Wednesday, Thursday, Tuesday, and 2nd Wednesday for CO and Tuesday and 2nd Wednesday for NO and NOx. Consequently, emission factors have been calculated by regressing CO and NOx data against CO_2_ in three ways: regressing all weekday and weekend data, regressing weekday data, and regressing all weekday data where complete 24 h data sets are available. Figure 6 and Figure 7 show data and regression lines for CO and NOx, respectively. Emission factors are calculated from the slopes of these lines; for CO slopes range from 0.0129 to 0.0138 (ppm CO)/ppm CO_2_) with r^2^ values ranging from 0.64 to 0.68, for NOx the slopes range from 0.66 to 0.80 (ppb NO/ppm CO_2_) with r^2^ values ranging from 0.62 to 0.73.

These regression results have been converted to fuel specific emissions (g/kg_fuel_) and regulated emissions (g/mile) in Table 4 below. Additional information included are the upper and lower confidence limits for all data regressions and the estimated uncertainty of the measurement. The latter is based on the range of the slopes and an estimated measurement uncertainty of 20% for the low-cost sensors used here.

In terms of fuel specific emissions, CO ranged from 26 to 28 (g/kg_fuel_) with and estimated uncertainty of 6 (g/kg_fuel_). Other reported emission factors range from 17 to 62 (g/kg_fuel_) [63,64] depending on the car manufacturing year.

For NOx the range is from 2.1 to 2.7 (g/kg_fuel_) with an estimated uncertainty of 0.5 (g/kg_fuel_). Our measurements showed very little correlation between NO_2_ and CO_2_ so nearly all of the variation of NOx with CO_2_ shown in the plots is due to NO. Regulated emissions of NOx are calculated following EPA regulations from the sum of NO and NO_2_ but assuming the molecular mass of NO_2_ for both species. On the other hand, fuel specific emissions of NO have been calculated based on the molecular mass of NO. This gives NO emissions ranging from 1.3 to 1.7 (g/kg_fuel_) with an estimated uncertainty of 0.3 (g/kg_fuel_). Other reported fuel specific emissions for NO range from 1.44 to 10.8 (g/kg_fuel_) [63,64,65] depending on the car manufacturing year.

We did not attempt to calculate fuel specific emissions of NO_2_ because of its low correlation with CO_2_ (r^2^ < 0.05). However, others have reported from 0.16 to 1.2 (g/kg_fuel_) [63,64,65] depending on the car manufacturing year.

Emissions of CO and NOx in g/mile have been calculated for comparison with Federal vehicle emission standards for 2017 are also shown in Table 4. For CO regulated emissions range from 2.4 to 2.9 g/mile with an uncertainty of 0.6 g/mile compared to the 2017 standard of 1.7 g/mile. The calculated NOx emissions are much higher relative to the Federal standard, ranging from 0.19 to 0.25 g/mile with an uncertainty of 0.05 g/mile compared to the 2017 standard of 0.03 g/mile. Emissions of NO are not regulated directly but are usually the main contributor the NOx emissions.

## 4. Conclusions

The daily average concentrations of CO and NO are much higher in the parking garage than the ambient levels measured in a nearby AMS and clearly related to vehicle traffic. Concentrations are more closely correlated with exit traffic than entry traffic. This is because there are higher emissions from vehicles warming up. Concentrations are in the mid-range, compared to other measurements conducted in underground parking garages. Concentrations of NO_2_, and PM2.5 were somewhat above ambient levels and weakly associated with traffic. Compared to other work NO_2_ concentrations in this study were in the mid-range but PM2.5 concentrations were much lower. Ozone concentrations in the garage were in the same range as ambient and inversely related to traffic due to titration by NOx. Concentrations of LDSA, a measure of ultrafine particles, were correlated with traffic and even more strongly correlated with CO and NO the two primary gaseous tailpipe emissions measured. No background concentrations of LDSA were available but the concentration range was similar roadside measurements made elsewhere.

Of the pollutants measured, there are EPA ambient standards for only CO, NO_2_, O_3_, and PM2.5. In all cases the levels in the parking garage are well below the standards.

In terms of fuel specific emissions (g/kgfuel), CO and NO values are lower than other reported in measurements in parking garages. Estimated regulated emissions (g/mile) of CO and, especially, NOx are above Federal vehicle emission standards for 2017.

The results obtained by the NO_2_ and PM2.5 sensors must be considered with caution. This is because of the relatively low R^2^ of the NO_2_ during calibration. The correlation between the SidePak PM2.5 measurements and all the other sensors was 0.5 or less.

This work has provided useful information of air quality in an underground parking garage, but it has limitations and further work is suggested. For example, the O_3_ and NO_2_ sensors behaved badly during the first two days of the study. Data retrieval problem with CO_2_ and NO sensors resulted in loss about 21% of data. The OPC-N2 PM2.5 sensor behaved poorly during the calibration and for that reason was not used during this study. Low-cost sensors continue to improve and further work with new suite of the sensors is recommended.

This was a short-term pilot study intended to demonstrate the use of low-cost sensors in confined spaces exposed to vehicle traffic. These measurements were conducted during summer 2019, longer term summer measurements as well as similar studies during winter, fall, and spring are recommended. It would be also useful locate the sensors in different locations within the parking garage. Calibration drift was a potential issue in this study in more frequent calibration are recommended in future studies. Further research is also needed to evaluate the impact of the heavy traffic associated with a major event such as football game or concert.

## Figures and Tables

**Figure 1 ijerph-19-15223-f001:**
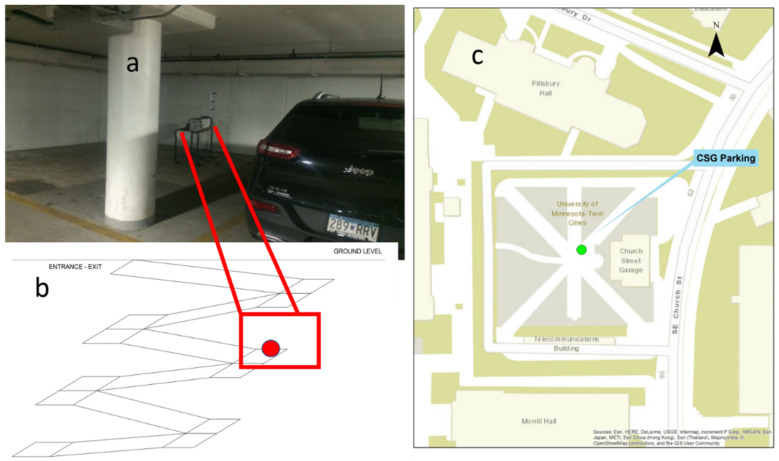
(**a**) location of the measurements in the parking garage. (**b**) schematic of the parking garage and the specific location of the MAAQSBox. (**c**) location of the parking garage in East bank campus at University of Minnesota.

**Figure 2 ijerph-19-15223-f002:**
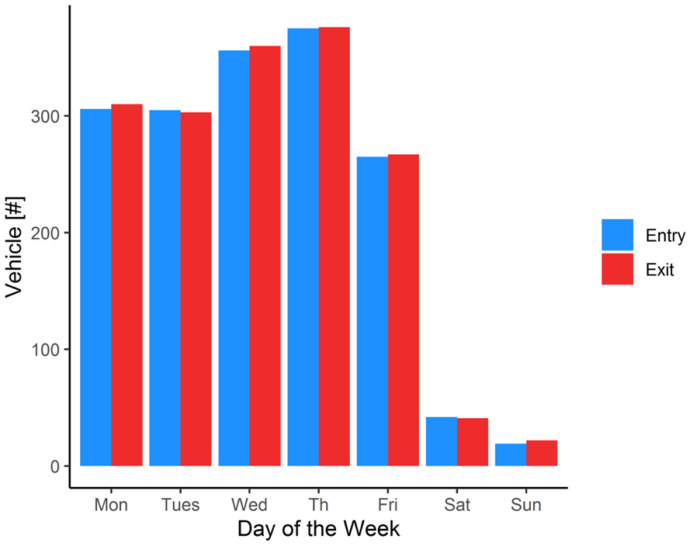
Average entry and exit of number of vehicles per day by day of the week in the parking garage in East bank campus at University of Minnesota.

**Figure 3 ijerph-19-15223-f003:**
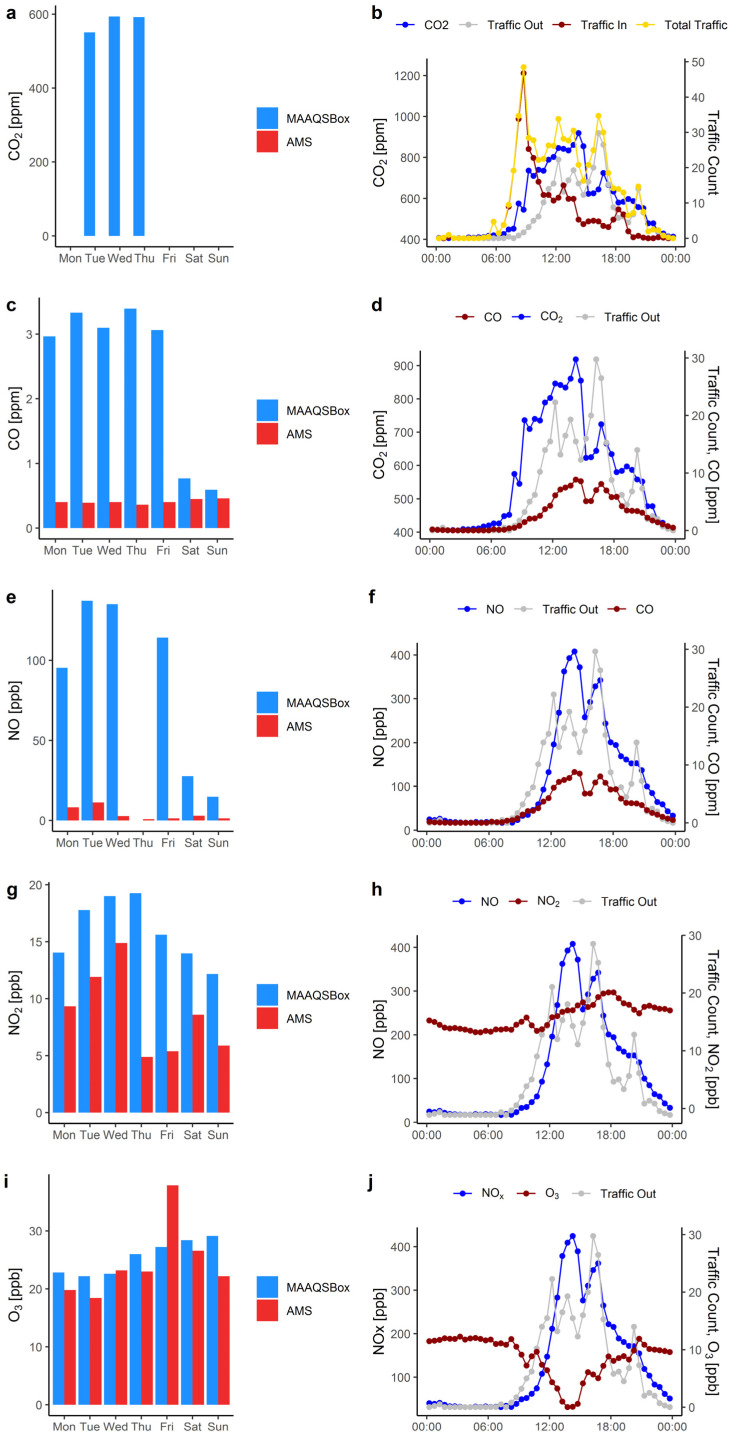
(**a**) Daily CO_2_ Concentrations. (**b**) daily variation of entry, exit, and total traffic, and CO_2_ concentration with time of day, weekday 30 min averages. (**c**) Daily average CO concentrations measured in the garage by the MAAQSBox and ambient levels measured at the nearest AMS and (**d**) daily variation of CO and CO_2_ concentration and exit traffic, weekday 30 min averages. (**e**) Daily average NO concentrations measured in the garage by the MAAQSBox and ambient levels measured at the nearest AMS and (**f**) daily variation of NO and CO concentration and exit traffic, weekday 30 min averages. (**g**) Daily average NO_2_ concentrations measured in the garage by the MAAQSBox and ambient levels measured at the nearest AMS and (**h**) daily variation of NO_2_ and NO concentration and exit traffic, weekday 30 min averages. (**i**) Daily (**a**) Daily average O_3_ concentrations measured in the garage by the MAAQSBox and ambient levels measured at the nearest AMS and (**j**) daily variation of O_3_ and NOx concentration and exit traffic, weekday 30 min averages.

**Figure 4 ijerph-19-15223-f004:**
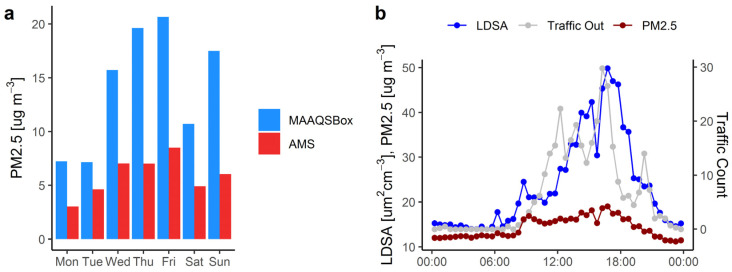
(**a**) Daily average PM2.5 concentrations measured in the garage by the MAAQSBox and ambient levels measured at the nearest AMS and (**b**) daily variation of PM2.5 and LDSA concentration and exit traffic, weekday 30 min averages.

**Figure 5 ijerph-19-15223-f005:**
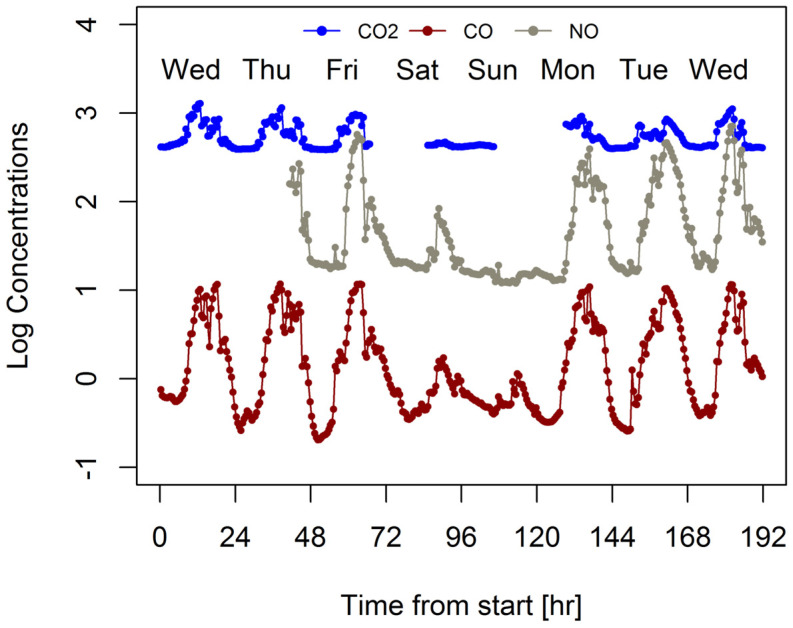
30 min average CO_2_, NO and CO plotted against time from start of test.

**Figure 6 ijerph-19-15223-f006:**
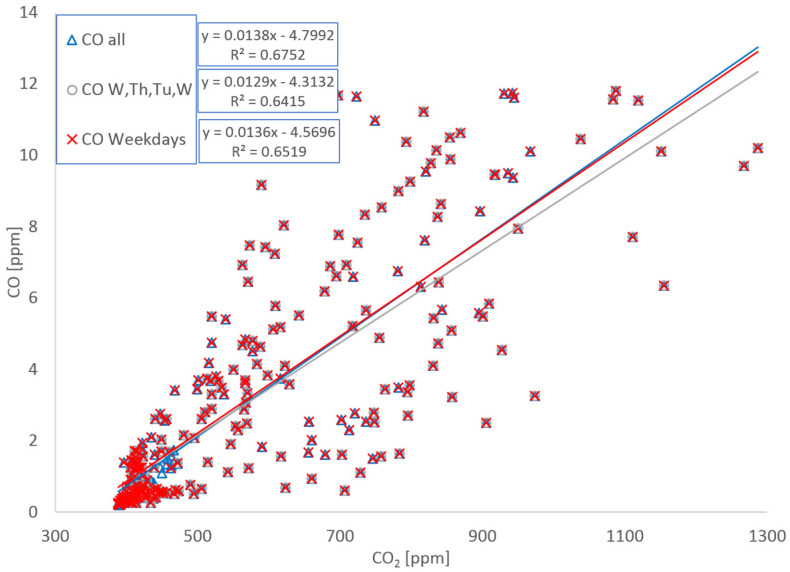
CO plotted against CO_2_ for three data sets: all data, all weekday data, all complete 24 h data sets—in this case Wed, Thu, Tue, and 2nd Wed.

**Figure 7 ijerph-19-15223-f007:**
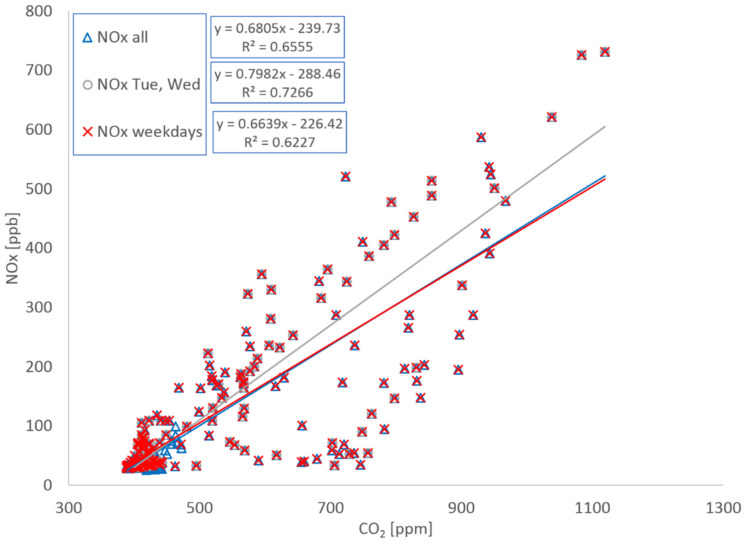
NOx plotted against CO_2_ for three data sets: all data, all weekday data, all complete 24 h data sets—in this case Tue and 2nd Wed.

**Table 1 ijerph-19-15223-t001:** Calibration results of B4 gas sensors. N represents the number of hours of calibration for each sensor in the AMS. R^2^ is the coefficient of determination of MLR. SE is the standard deviation of mean. RC represents the range of the concentration during the calibration.

Sensor	N	R^2^	SE	RC
CO	515 h	0.95	0.04 ppm	0.25 ppm–1.1 ppm
NO	509 h	0.79	3.1 ppb	1.3 ppb–46.4 ppb
NO_2_	413 h	0.63	3.2 ppb	1 ppb–28 ppb
O_3_	407 h	0.93	2.8 ppb	0 ppb–54 ppb

**Table 2 ijerph-19-15223-t002:** Overall correlations among variables in the study.

	CO ppm	CO_2_ ppm	NO ppb	NO_2_ ppb	NOx ppb	O_3_ ppb	LDSA µm^2^cm^−3^	PM2.5 µg m^−3^	Total Traffic In	Total Traffic Out
CO ppm	1.00									
CO_2_ ppm	0.82	1.00								
NO ppb	0.96	0.81	1.00							
NO_2_ ppb	0.47	0.21	0.41	1.00						
NOx	0.96	0.81	1.00	0.43	1.00					
O_3_ ppb	−0.72	−0.68	−0.66	−0.57	−0.67	1.00				
LDSA	0.75	0.60	0.75	0.50	0.75	−0.50	1.00			
PM2.5	0.27	0.36	0.12	0.02	0.11	0.14	0.48	1.00		
Total Traffic In	0.15	0.35	0.11	0.10	0.11	−0.22	0.14	0.09	1.00	
Total Traffic Out	0.72	0.59	0.66	0.36	0.66	−0.46	0.54	0.18	0.15	1.00

**Table 3 ijerph-19-15223-t003:** Garage and MPCA air monitoring station (AMS) 1 h average measurements.

	CO_2_	LDSA	CO	CO AMS	NO	NO AMS	NO_2_	NO_2_ AMS	O_3_	O_3_ AMS	PM2.5	PM2.5 AMS
	ppm	μm^2^/cm^3^	ppm	ppm	ppb	ppb	ppb	ppb	ppb	ppb	mg/m^3^	mg/m^3^
maximum	1267	98.5	11.7	0.7	518.4	26.6	24.5	20.0	35.4	52.0	35.5	22.0
minimum	389	5.8	0.2	0.3	12.3	0.1	8.4	1.0	7.2	1.0	4.2	0.0
average	564	22.0	2.5	0.4	80.1	3.9	15.4	7.7	25.9	26.3	14.9	6.1
stdev	188	13.2	3.0	0.1	111.7	5.7	3.6	4.5	5.1	11.6	7.1	3.4
count	128	168	168	168	127	168	144	168	144	168	168	168

**Table 4 ijerph-19-15223-t004:** Fuel specific emissions for CO, NOx and NO and regulated emissions for CO and NOx. Range of estimates and confidence limits shown.

**CO**	**All**	**Lower 95.0%**	**Upper 95.0%**	**Weekdays**	**W, Th, Tu, W**	**Uncert ±**
CO/CO_2_ slope	0.014	0.013	0.015	0.014	0.013	
CO/CO_2_ g/kg	8.8	8.1	9.5	8.7	8.2	1.9
CO g/kgf	28.1	25.9	30.3	27.6	26.2	6.0
g/km	1.7	1.5	1.8	1.6	1.5	0.4
g/mi	2.6	2.4	2.9	2.6	2.5	0.6
2017 std g/mi	1.7	1.7	1.7	1.7	1.7	
**NOx**	**All**	**Lower 95.0%**	**Upper 95.0%**	**Weekdays**	**Tu, W**	**Uncert ±**
NOx/CO_2_ slope	0.68	0.61	0.75	0.66	0.80	
NOx/CO_2_ g/kg	0.71	0.64	0.78	0.69	0.83	0.16
g/kgf	2.3	2.1	2.5	2.2	2.7	0.50
g/km	0.13	0.12	0.15	0.13	0.16	0.03
g/mi	0.21	0.19	0.23	0.21	0.25	0.05
2017 std	0.03	0.03	0.03	0.03	0.03	

Assumptions: 30 mpg, 0.75 kg_fuel_/L, 7.84 L/100 km, 0.0588 kg_fuel_/km.

## Data Availability

The data are available from Andres Gonzalez (gonza817@umn.edu).

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
