# Peer review of "Measuring the Air Quality Using Low-Cost Air Sensors in a Parking Garage at University of Minnesota, USA"

_ijerph, 2022, doi:10.3390/ijerph192215223_

Round 1
Reviewer 1 Report
This paper addressed the concentration of air pollutants in underground parking garages and the ambient air. However, in this paper, the authors focus more on descriptions of pollutant monitoring concentrations with less novelty. Authors should pay more attentions to the academic enlightenment of this study to the relevant research field. There are several crucial issues need to be reorganized as follows:
Line 28: How you raise the research questions? More critical and detailed research background introductions are needed. Besides, authors should address the meaning of using low-cost sensors (one of key words) in the introduction.
Line 95: More than one MAAQSbox would be better to measure air concentrations in this case? Explain it.
Line 124: (Table 1 N)?
Line 157: A significant test, or confidence interval, of the correlation coefficient is required.
Line 165: “#”?
Line 171: The framework of the table is lacking.
Line 234: In this study, low concentration of ozone may limit the oxidation of NO to NO2. It is suggested to make some comparisons between here and outdoor parking lots to see if low concentration of ozone is a limit to NO2 generation
Line 305: The sense of the right accessory coordinate is discontinuous in Fig d,f,h, j. . Moreover, the values of“traffic out” sometimes turned negative and sometimes positive, and the accessory coordinate also contains a negative concentration, which does not match “traffic out” with confusing readers. It is suggested to take out traffic out and display it in the blank area of these images in the form of subgraph. Finally, for fig j, this figure presents the daily variation of NOx concentration, but the trend of the curve is similar to the daily variation of NO concentration in Fig f and h. The authors need to check the NOx concentration data.
Line 325: These formulas and related descriptions can be placed in the Methods section.
Line 328: The data source for kg C and kg Fuel?
Line 389: Title of table?
Line 393: More further discussions about the impacts of related pollution gas on human health are needed. Moreover, more limitation should be mentioned and the future work related to the results should be outlined too.
Line 443: More updated literature is needed.
Reviewer 2 Report
In the article the results of mesurements of CO, NO, NO2, PM2.5 and LDSA in underground garage at University of Minnesota were presented. Results of mesurement made underground were compared with results of air monitoring system located outdoor. The content of the article is interesting, but the presentation of the measurement results requires improvement. Below are some suggestions to improve the quality of the article:
- At point 2.1 the location of AMS at figure 1 schould be adeed;
- At point 2.2 the accurancy of used sensors schould be added;
- In article the large number of abbreviations is ussed (N, SE, RC, AMS, CSG ........) it is recomendet to add a list of abbreviations at the beggining of article;
- What is the premisible level of CO, NO2, O3 and PM2,5 by EPA;
- At table 3 thera are a negative values of concentration PM 2.5 this result is non-physical and should not be taken into account;
- Figure 3 should be cut into smaller figures and insert it in the text, this will increase the readability of the article;
- At figure 3 the upper and lower indexes schould be aded;
- Why at figure 3 d, f and l the trafic count were differ from each other, it is recomended to schow the results of all mesurements from one day;
- Why at figures 3 h-J the trafic count have the negative value ?
- How were calculated "CO2 added by trafic " were calculated ? The formula for calculations of emmision coeficients schould be described.
Best regards for authors.
Reviewer 3 Report
The paper is well-written and has a significant contribution towards seeing the effect of parking-related pollution.
1. The authors have not mentioned anywhere the LOD of the instruments used in this study.
2. Have the authors collected any meteorological observation data inside the parking?
3. At the time of entry and exit, the wind speed becomes high which disperses the pollutants so fast. Can authors point out this concern?
4. Did the authors observe any spikes in the data during the entry and exit time of the vehicles?
5. What was the location of the instrument, at which monitoring data was collected? Does instrument location affect the observations?
Round 2
Reviewer 1 Report
The authors have responsed to the questions.